# ProtoVAE: A Trustworthy Self-Explainable Prototypical Variational Model

**Srishti Gautam**[1], **Ahcene Boubekki**[1], **Stine Hansen**[1], **Suaiba Amina Salahuddin**[1]
**Robert Jenssen**[1], **Marina MC Höhne**[2,1], **Michael Kampffmeyer**[1]
[1]UiT The Arctic University of Norway
[2]Technical University of Berlin

## Abstract

The need for interpretable models has fostered the development of self-explainable classifiers. Prior approaches are either based on multi-stage optimization schemes, impacting the predictive performance of the model, or produce explanations that are not transparent, trustworthy or do not capture the diversity of the data. To address these shortcomings, we propose ProtoVAE, a variational autoencoder-based framework that learns class-specific prototypes in an end-to-end manner and enforces *trustworthiness* and *diversity* by regularizing the representation space and introducing an orthonormality constraint. Finally, the model is designed to be *transparent* by directly incorporating the prototypes into the decision process. Extensive comparisons with previous self-explainable approaches demonstrate the superiority of ProtoVAE, highlighting its ability to generate trustworthy and diverse explanations, while not degrading predictive performance.

## 1 Introduction

Despite the substantial performance of deep learning models in solving various automated real-world problems, lack of transparency still remains a crucial point of concern. The black-box nature of these high-accuracy achieving models is a roadblock in critical domains such as healthcare [1, 2], law [3], or autonomous driving [4]. This has led to the emergence of the field of explainable artificial intelligence (XAI) which aims to justify or explain a model's prediction in order to increase trustworthiness, fairness, and safeness in the application of the complex models henceforward.

Consequently, two lines of research have emerged within XAI. On the one hand, there are general methodologies explaining a posteriori black-box models, so-called post-hoc explanation methods [5, 6, 7]. While on the other hand, there are models developed to provide explanations along with their predictions [8, 9, 10]. The latter class of models, also known as self-explainable models (SEMs), are the focus of this work. Recently, many methods have been developed for quantifying post-hoc explanations [11]. However, there is still a lack of a concise definition of what SEMs should encompass, thus a lack of comparability of recent methods [12].

Methodologically, a large number of SEMs follow the approach of concept learning, analogous to prototype or basis feature learning, where a set of class representative features are learned [8, 9]. In this paper we gauge SEMs through the prism of three properties. First and foremost, the prototypes should be visualizable in the input space, and these transparent concepts should directly be employed by a glass-box classification model. Many of the existing approaches try to imitate prototype transparency by using nearest training samples to visualize the prototypes [8, 13], while some flatly use training images as prototypes preventing an end-to-end optimization and limiting the flexibility of the model [9, 14]. Secondly, the prototypes should exhibit both inter-class and intra-class diversity. Methods failing to ensure this property [9] are prone to prototype collapse into a single point which necessarily undermines their performance. Finally, SEMs should perform comparable to their black-

box counterparts while producing robust and faithful explanations. Previous approaches have a tendency to achieve self-explanability by sacrificing the predictive performance [9, 14, 13].

To address the aforementioned shortcomings of current SEMs, we introduce ProtoVAE, a prototypical self-explainable model based on a variational autoencoder (VAE) backbone. The architecture and the loss function are designed to produce *transparent*, *diverse*, and *trustworthy* predictions, as well as explanations, while relying on an end-to-end optimization. The predictions are linear combinations of distance-based similarity scores with respect to the prototypes in the feature space. The encoder and decoder are trained as a mixture of VAEs sharing the same network but each with its own Gaussian prior centered on one of the prototypes. The latter are enjoined to capture diverse characteristics of the data through a class-wise orthonormality constrain. Consequently, our learned prototypes are truly transparent global explanations that can be decoded and visualized in the input space. Further, we are able to generate local pixel-wise explanations by back-propagating relevances from the similarity scores. Empirically, our model corroborates trustworthiness both in terms of performance as well as the quality of its explanations.

Our main contributions can be summarized as follows:
- We define three properties for SEMs, based on which we present a novel prototypical self-explainable model with a variational auto-encoder backbone, equipped with a fully *transparent* prototypical space.
- We are able to learn faithful and *diverse* global explanations easily visualizable in the input space.
- We provide an extensive qualitative and quantitative analysis on five image classification datasets, demonstrating the efficiency and *trustworthiness* of our proposed method.

## 2    Predicates for a self-explainable model

For the benefit of an efficient and comprehensible formalization of SEMs, we here define three properties that we consider as prerequisites for SEMs.

**Definition 1** *An SEM is **transparent** if:*
   *(i) its concepts are utilized to perform the downstream task without leveraging a complex black-box model;*
   *(ii) its concepts are visualizable in input space.*

**Definition 2** *An SEM is **diverse** if its concepts represent non-overlapping information in the latent space.*

**Definition 3** *An SEM is **trustworthy** if:*
   *(i) the performance matches to that of the closest black-box counterpart;*
   *(ii) the explanations are robust, i.e., similar images yield similar explanations.*
   *(iii) the explanations represent the real contribution of the input features to the prediction.*

Note that these definitions echo properties and axioms found in other works. However, the view of such properties is diverse across the literature which leads to failure of encompassing the wide research of SEMs in general. For example, *transparency* is known as 'completeness' in [15] and 'local accuracy' in [16]. In the next section, we provide a comparison of existing SEMs based on the fulfillment of the proposed predicates.

## 3    Categorization of related self-explainable works

Self-explainable models optimize for both explainability and prediction, making the network inherently interpretable. As our main contribution is a prototypical model, we review and categorise existing prototypical SEMs according to the above-mentioned properties.

SENN [8] introduces a general self-explainable neural network designed in stages to behave locally like a linear model. The model generates interpretable concepts, to which sample similarities are directly aggregated to produce predictions. This generalized approach has been followed by most of the prototypical and concept-based self-explainable methods, and is also mirrored by our approach. SENN, however uses training data to provide interpretation of learned concepts, therefore approximating transparency, unlike our model which by-design has a decoder to visualize prototypes.

Table 1: Summary of the SEM properties satisfied by the baselines. The optimization scheme is also indicated. The symbol $\sim$ indicates that the concepts cannot be directly visualized in the input space and that the nearest training data serve as ersatz.

| | Transparency | Diversity | Trustworthiness | Optimization |
|---|:---:|:---:|:---:|:---:|
| SENN[8] | $\sim$ | ✓ | ✓ | End-to-end |
| ProtoPNet[9] | ✓ | | | Alternating |
| TesNET[14] | ✓ | ✓ | | Alternating |
| SITE[17] | | ✓ | ✓ | End-to-end |
| FLINT[13] | $\sim$ | ✓ | | End-to-end |
| ProtoVAE | ✓ | ✓ | ✓ | End-to-end |

ProtoPNet [9] is a representative of a line of works [9, 14, 18, 19, 20], where a prototypical layer is introduced before the final classification layer. For maintaining interpretability, the prototypes are set as the projection of closest training image patches after every few iterations during training. Our method is closely related to ProtoPNet with the distinction of decode-able learned prototypes yielding a smooth and regularized prototypical space, thus allowing more flexibility in the model. TesNet [14] extends ProtoPNet and improves diversity at class level using five loss terms. Similarly to our approach, they distribute the base concepts among the classes and include an orthonormal constraint. However, the basis concepts are still projections from the nearest image patches, which leads to loss in predictive performance, similar to ProtoPNet. SITE [17] generates class prototypes from the input and introduces a transformation-equivariant model by constraining the interpretations before and after transformation. Since the prototypes are dynamic and generated for each test image, this method only provides local interpretations and lacks global interpretations. FLINT [13] introduces an interpreter model (FLINT-$g$) in addition to the original predictor model (FLINT-$f$). Although FLINT-$f$ has been introduced by the authors as a framework that learns in parallel to the interpretations, it is not an SEM on its own. Therefore, we focus on FLINT-$g$, henceforward referred to as FLINT. FLINT takes as input features of several hidden layers of the predictor to learn a dictionary of attributes. However, the interpreter is not able to approximate the predictor model perfectly, therefore losing *trustworthiness*. Unlike prior approaches, ProtoVAE is designed to fulfill all three SEM properties. We summarize the discussed methods and their categorization in Table 1.

## 4 ProtoVAE

In this section, we introduce ProtoVAE, which is designed to obey the aforementioned SEM properties. Specifically, *transparency* is in-built in the architecture and further enforced along with *diversity* through the loss function. Also, we describe how our choices ensure the *trustworthiness* of our method.

### 4.1 Transparent architecture

In a *transparent* self-explainable model, the predictions are interpretable functions of concepts visualizable in the input space. To satisfy this property, we rely on an autoencoder-based architecture as backbone and a linear classifier. In order to have consistent, *robust*, and *diverse* global explanations, we consider prototypes in a greater number than classes. Unlike previous prototypical methods [9, 14], which update the prototypes every few iterations with the embeddings of the closest training images, ProtoVAE is trained *end-to-end* to learn both the prototypes in the feature space and the projection back to the input space. This gives ProtoVAE the flexibility to capture more general class characteristics. To further alleviate situations where some of the optimized prototypes are positioned far from the training data in the feature space, possibly causing poor reconstructions and interpretations, we leverage a variational autoencoder (VAE). VAEs are known to learn more robust embeddings and thus generate better reconstructions from out-of-distribution samples than simple autoencoders [21]. A schematic representation of the network is depicted in Fig. 1.

**Details of the operations** The downstream task at hand is the classification into $K > 0$ classes of the image dataset $\mathcal{X} = \{(\boldsymbol{x}_i, \boldsymbol{y}_i)\}_{i=1}^N$, where $\boldsymbol{x}_i \in \mathbb{R}^p$ is an image and the one-hot vector $\boldsymbol{y}_i \in \{0,1\}^K$

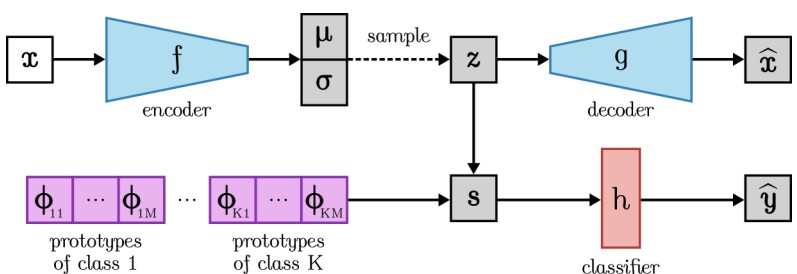

Figure 1: Schematic representation of the architecture of ProtoVAE. The input image $x$ is encoded by $f$ into a tuple $(\boldsymbol{\mu}, \boldsymbol{\sigma})$. A vector $z$ is sampled from $\mathcal{N}(\boldsymbol{\mu}, \boldsymbol{\sigma})$ which, on one side, is decoded by $g$ into the reconstructed input $\hat{x}$ and, on the other side, is compared to the prototypes $\phi_{kj}$ resulting in the similarity scores $s$. The latter are passed through the classifier $h$ to get the final prediction $\hat{y}$.

encodes its label. The network consists of an encoder $f : \mathbb{R}^p \to \mathbb{R}^d \times \mathbb{R}^d$, a decoder $g : \mathbb{R}^d \to \mathbb{R}^p$ ($d < p$), $M$ prototypes per class $\boldsymbol{\Phi} = \{\phi_{kj}\}_{j=1 \ldots M}^{k=1 \ldots K}$, a similarity function $\mathrm{sim} : \mathbb{R}^d \to \mathbb{R}^M$ and a glass-box linear classifier $h : \mathbb{R}^M \to [0,1]^K$. An image $x_i$ is first transformed by $f$ into a tuple $(\boldsymbol{\mu}_i, \boldsymbol{\sigma}_i) = f(x_i)$ which, in the VAE realm, are the parameters of the posterior distribution. A feature vector $z_i$ is then sampled from the normal distribution $\mathcal{N}(\boldsymbol{\mu}_i, \boldsymbol{\sigma}_i)$ and is used twice. On the one hand, it is decoded as $\hat{x}_i = g(z_i)$. On the other hand, it is compared to the prototypes. We use the same similarity function as in [9] and obtain the resulting vector $s_i \in \mathbb{R}^{K \times M}$ as:

$$s_i(k, j) = \mathrm{sim}(z_i, \phi_{kj}) = \log \left( \frac{||z_i - \phi_{kj}||^2 + 1}{||z_i - \phi_{kj}||^2 + \epsilon} \right), \tag{1}$$

with $0 < \epsilon < 1$. Finally, $s_i$ is used to compute the predictions: $\hat{y}_i = h(s_i)$. Moreover, the similarity vector $s_i$ captures the distance to the prototypes but also indicates the influence of each prototype on the prediction.

## 4.2 Diversity and trustworthiness

Unlike transparency, diversity cannot be achieved solely through the architectural choices. It needs to be further enforced during the optimization. Our architecture implies two loss terms: a classification loss and a VAE-loss. Without further regulation, our model is left vulnerable to the curse of prototype collapse [14, 22] which would undermine the SEM *diversity* property. We prevent such a situation with a third loss term enforcing orthonormality between prototypes of the same class. The loss function of ProtoVAE can thus be stated as follows:

$$\mathcal{L}_{\mathrm{ProtoVAE}} = \mathcal{L}_{\mathrm{pred}} + \mathcal{L}_{\mathrm{orth}} + \mathcal{L}_{\mathrm{VAE}}. \tag{2}$$

We detail now each term and discuss how they favor *diversity* and *trustworthiness*.

**Inter-class diversity through classification**   Although the prototypes are assigned to a class, the classifier is blind to that information. Thus, the prediction problem is a classic classification that we solve using the cross-entropy loss.

$$\mathcal{L}_{\mathrm{pred}} = \frac{1}{N} \sum_{i=1}^{N} \mathbf{CE}(h(s_i); y_i). \tag{3}$$

Since $h$ is linear, the loss pushes the embedding of each class to be linearly separable, yielding a greater *inter-class diversity* of the prototypes.

**Intra-class diversity through orthonormalization**   The inter-class diversity is guaranteed by the previous terms. However, without further regularization, the prototypes might collapse to the center of the class, obviating the possibilities offered by the extra prototypes. To prevent such a situation and foster intra-class diversity, we enforce the prototype of each class to be orthonormal to each other as follows:

$$\mathcal{L}_{\mathrm{orth}} = \sum_{k=1}^{K} ||\bar{\boldsymbol{\Phi}}_k^T \bar{\boldsymbol{\Phi}}_k - \boldsymbol{I}_M||_F^2, \tag{4}$$

where $\boldsymbol{I}_M$ is the identity matrix of dimension $M \times M$ and the column-vectors of matrix $\bar{\boldsymbol{\Phi}}_k$ are the prototypes assigned to class $k$ minus their mean, i.e., $\bar{\boldsymbol{\Phi}}_k = \{\phi_{kj} - \bar{\phi}_k, \; j = 1 \; .. \; M\}$ with $\bar{\phi}_k = \sum_{l=1}^{M} \phi_{kl}$. Beyond regularizing the Frobenius norm $||.||_F$ of the prototype, this term favors the disentanglement of the captured concepts within each class, which is one way to obtain *intra-class diversity*.

**Robust classification and reconstruction through VAE**  The VAE architecture ensures the robustness of the embedding and of the decoder. In its original form, the VAE loss considers a single standard normal distribution as a prior and is trained to minimize:

$$||\boldsymbol{x} - \hat{\boldsymbol{x}}||^2 + D_{\mathrm{KL}}\big(p_f(\boldsymbol{z}|\boldsymbol{x})||\mathcal{N}(\mathbf{0}_d, \mathbf{I}_d)\big), \tag{5}$$

where $\boldsymbol{I}_d$ is the identity matrix of dimension $d \times d$. Such an objective enjoins the embedding to organize as if generated by a single Gaussian distribution, thus making it difficult to split it with the linear classifier $h$. To help the classifier, we consider instead a mixture of VAEs sharing the same network each with a Gaussian prior centered on one of the prototypes. Since, each prototype has a label, only data-points sharing that label are involved in the training of the associated VAE. The loss function of our mixture of VAEs is (derivation in the supplementary material Sec. S2):

$$\mathcal{L}_{\mathrm{VAE}} = \frac{1}{N} \sum_{i=1}^{N} ||\boldsymbol{x}_i - \hat{\boldsymbol{x}}_i||^2 + \sum_{k=1}^{K} \sum_{j=1}^{M} \boldsymbol{y}_i(k) \frac{\boldsymbol{s}_i(k,j)}{\sum_{l=1}^{M} \boldsymbol{s}_i(k,l)} D_{\mathrm{KL}}\big(\mathcal{N}(\boldsymbol{\mu}_i, \boldsymbol{\sigma}_i)||\mathcal{N}(\phi_{kj}, \mathbf{I}_d)\big). \tag{6}$$

In addition to training the decoder, this loss enjoins the embedding to gather closely around their class prototypes.

### 4.3  Visualization of explanations

ProtoVAE is designed to have the inherent capability to reconstruct prototypes via the decoder, which is trained to approximate the input distribution. Additionally, to generate faithful pixel-wise local explanation maps, we build upon the concepts of Layer-wise relevance propagation (LRP) [23] which is a model-aware XAI method computing relevances based on the contribution of a neuron to the prediction. Following [12], we generate explanation maps, where for each prototype, the similarity of an input to the prototype is backpropagated to the input image according to the LRP rules. For an input image $\boldsymbol{x}_i$, the point-wise similarity between the transformed mean vector $\boldsymbol{\mu}_i$ with a prototype $\phi_{kj}$ is first calculated as:

$$\boldsymbol{\gamma}_{ikj} = \frac{1}{\boldsymbol{d}_{ikj} + \eta} \quad \text{with} \quad \boldsymbol{d}_{ikj} = (\boldsymbol{\mu}_i - \phi_{kj}) * (\boldsymbol{\mu}_i - \phi_{kj}), \tag{7}$$

where $*$ is the Hadamard element-wise product and $\eta > 0$. The similarity $\boldsymbol{\gamma}_{ikj}$ is then backpropagated through the encoder following LRP composite rule, which is known as best practice [24] to compute local explanation maps. Following this, the $\mathrm{LRP}_{\alpha\beta}$ rule is applied to the convolutional layers and the Deep Taylor Decomposition based rule $\mathrm{DTD}_{z^B}$ [25] is applied to the input features.

## 5  Experiments

In this section, we conduct extensive experiments to evaluate ProtoVAE's trustworthiness, transparency, and ability to capture the diversity in the data. More specifically, we demonstrate the trustworthiness of our model in terms of predictive performance in Sec. 5.1. Qualitative evaluations are then conducted in Sec. 5.2 to verify the diversity and transparency properties, followed by a quantitative evaluation of the explanations corroborating its trustworthiness. Additionally, we provide an ablation study for the terms in Eq. 2 and further study the effect of the L2 norm in Eq. 6 on the prototype reconstructions in the supplementary material in Sec. S6.1 and Sec. S6.9, respectively.

**Datasets and implementation:**  We evaluate ProtoVAE on 5 datasets, MNIST [26], FashionMNIST [27] (fMNIST), CIFAR-10, [28], a subset of QuickDraw [29] and SVHN [30]. We use small encoder networks with 4 convolution layers for MNIST, fMNIST and CIFAR-10, 3 for QuickDraw and 8 for SVHN. These convolution layers are followed by 2 linear layers which gives us the tuple $(\boldsymbol{\mu}_i, \boldsymbol{\sigma}_i)$ for each image $i$. The decoder mirrors the encoder's architecture. Similar to [9], we fix

Table 2: Performance results of ProtoVAE compared to other state-of-the-art methods (measured in accuracy (in %)). The reported numbers are means and standard deviations over 4 runs. Best and statistically non-significantly different results are marked in bold. *Results for SITE are taken from the original paper and thus based on more complex architectures.

|  | Black-box encoder | FLINT [13] | SENN [8] | *SITE [17] | ProtoPNet [9] | ProtoVAE |
|---|---|---|---|---|---|---|
| MNIST | 99.2±0.1 | **99.4±0.1** | 98.8±0.7 | 98.8 | 94.7±0.6 | **99.4±0.1** |
| fMNIST | 91.5±0.2 | 91.5±0.2 | 88.3±0.3 | - | 85.4±0.6 | **91.9±0.2** |
| CIFAR-10 | 83.9±0.1 | 79.6±0.6 | 76.3±0.2 | 84.0 | 67.8±0.9 | **84.6±0.1** |
| QuickDraw | 86.7±0.4 | 82.6±1.4 | 79.3±0.3 | - | 58.7±0.0 | **87.5±0.1** |
| SVHN | **92.3±0.3** | 90.8±0.4 | 91.5±0.4 | - | 88.6±0.3 | **92.2±0.3** |

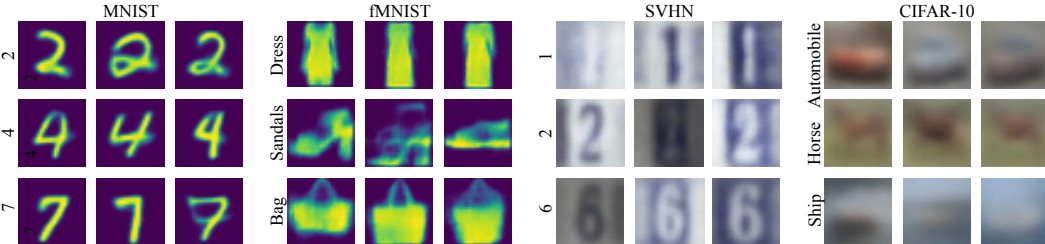

Figure 2: Visualization of learned prototypes for different classes for MNIST, fMNIST, SVHN and CIFAR-10.

the prototypes per class, $M$, to 5 for MNIST and SVHN and 10 for the other datasets. Further details about the datasets and additional implementation details, such as the detailed architecture and hyperparameters, are provided in the supplementary material Sec. S3 and S4. Our code is available at https://github.com/SrishtiGautam/ProtoVAE.

**Baselines** To ensure a fair comparison, we modified the publicly available code of ProtoPNet, FLINT and SENN to use the same backbone network as ProtoVAE and when relevant the same number of prototypes per class as used for ProtoVAE. We also provide the results for the predictive performance of SITE as reported in [17], since the code is not publicly available. We also report the performance of our model with a ResNet-18 backbone in the supplementary material Sec. S6.2. Further, we compare ProtoVAE using FLINT's encoders as provided in [13] for both FLINT and SENN in Sec. S6.3 in the supplementary material. Finally, we also compare with the black-box counterpart of our model, i.e, a classical feed-forward CNN based on the same encoder as ProtoVAE but followed by a linear classifier and trained end-to-end with the cross-entropy loss. This black-box encoder model is thus free from all regularization necessary for self-explainability.

## 5.1 Evaluation of predictive performance

In the Table 2, we can observe that ProtoVAE surpasses all other SEMs in terms of predictive performance on all five datasets, which is based on its increased flexibility in the architecture. For ProtoPNet, we observe a gap in performance, which is due to the low number of optimal class-representatives in the actual training data. This creates a huge bottleneck at the prototype layer and therefore limits its performance. Further, and more importantly, when compared to the true black-box counterpart, ProtoVAE achieves no loss in accuracy and is even able to perform better on all the datasets. We believe this is due to an efficient over-clustering of the latent space with the flexible prototypes, as well as the natural regularizations achieved through the VAE model. These results strengthens the *trustworthiness* of ProtoVAE in terms of the predictive performance.

## 5.2 Evaluation of explanations

**Qualitative evaluation** The demonstrated results in this section strengthen the fulfilment of the *transparency* property by providing human-understandable explanations for ProtoVAE. We visualize the decoded prototypes for different datasets, which act as global explanations for the corresponding

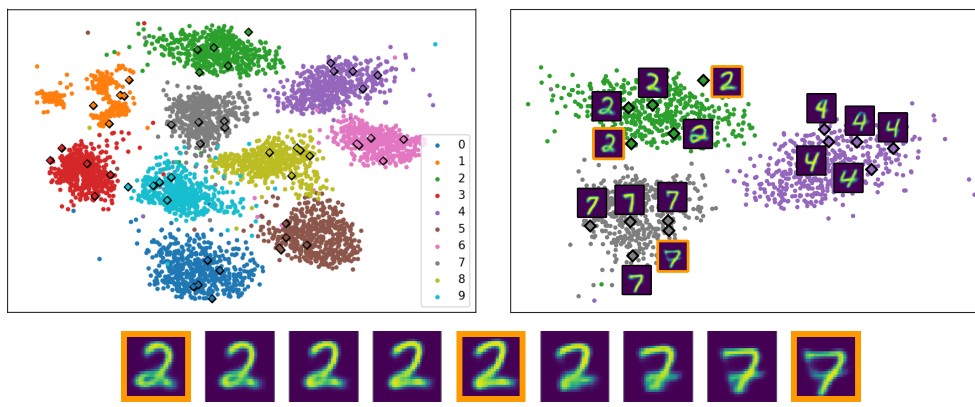

Figure 3: UMAP representations for the prototypical space for MNIST (left), decoded prototypes overlayed for classes 2, 4 and 7 (right), and interpolation between prototypes of the same class (2) and between prototypes of different classes (2-7) (bottom).

classes, in Fig. 2 [1]. The prototypes for MNIST demonstrate that class 2 consist of 2's with flat bottom line or with rounded bottom lines. For fMNIST, the sandals class consists of both heels and flats. The class prototypes thus directly help visualizing the components of the classes by looking at a fixed number of prototypes per class instead of all the training data. Interestingly, although SVHN often contains multiple digits of different classes in the same training image, our prototypes efficiently capture only one digit representing its class. Moreover, a blurring effect is observed in our prototypes which captures more variability and therefore suggests efficient representation of the true "mean" of a subset of a given class, as opposed to other methods [9, 14, 13] which show the closest training images and are therefore sharper. This behavior supports our claim of more flexibility in the network, therefore enhancing predictive capability along with the ability to provide more faithful explanations. This blurring effect is observed to be more prominent in CIFAR-10, which is due to the high complexity in each class in the dataset and can be reduced by using a larger number of prototypes per class.[2] Additionally, to provide more clear visualization of the learned transparent prototypical space, we show UMAP representations of the prototypes and the training data for MNIST in Fig. 3. This visualization further illustrates the inter-class as well as the intra-class *diversity* of the prototypes. Moreover, due to the regularized prototypical space, we are efficiently able to interpolate between prototypes both within a class and between classes, therefore making the latent space fully *transparent*. In Fig. 3, we interpolate between 2 different prototypes of class '2' and from a prototype of class '2' to a class '7' prototype.

The local explanability maps for a test image according to the three closest, i.e. most similar, prototypes for both ProtoVAE and ProtoPNet are shown in Fig. 4, along with the corresponding similarity scores. As seen, different prototypes of the same class activate different parts of the same test image, which therefore helps in achieving better performance. The ProtoPNet maps are extremely coarse which therefore makes them challenging to interpret. Therefore, we overlay the heatmaps over the input image for ProtoPNet. As observed, the most activated prototypes do not belong to the same class as the test image. This might happen because of ProtoPNet focusing on patches in prototypes, therefore losing contextual information. The 3 closest prototypes shown for the image 'apple' belong to class 'lion'. Further, an uninformative training image, which is not seen in the ProtoVAE prototypes, has been selected by ProtoPNet to represent 5 out of 10 prototypes for class 'lion'. The remaining 5 prototypes are represented by 1 other same training image. This effect is seen predominantly in ProtoPNet where the prototypes of the same class collapse to one point and are thus represented by the same training image, therefore dissatisfying the *diversity* property, as opposed to ProtoVAE. The prototypes for class 'lion' for both the models are included in the supplementary material in Sec. S6.4.

---

[1] As a reference to gauge the quality and sharpness of the pictures of Fig. 2, reconstructions of test images are provided in Sec. S6.11.

[2] We demonstrate this behavior in Sec. S6.8 and show in Sec. S6.7 how local explainability maps can be used to gather additional information about pixel-wise relevances thereby counterbalancing blurry prototypes.

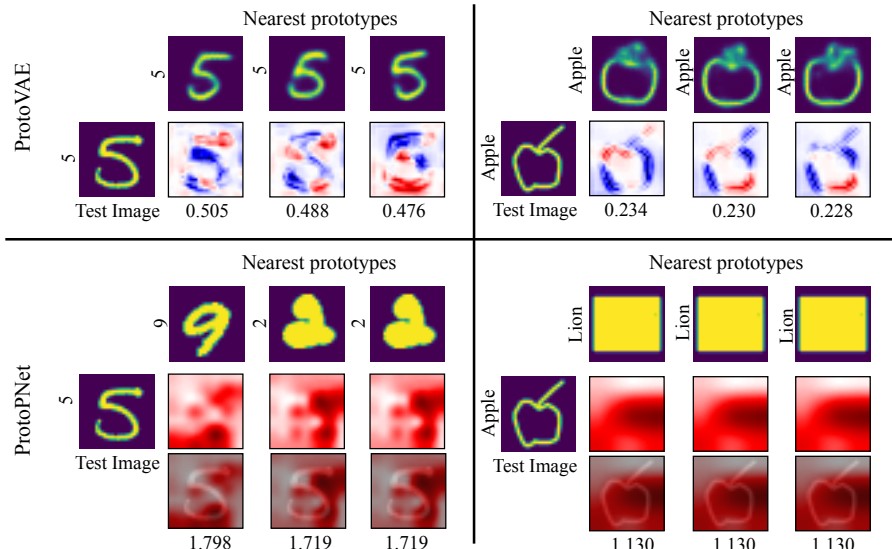

Figure 4: Three maximally activated prototypes, the corresponding prototypical activations, and corresponding similarity scores for a test image of class 5 (for MNIST) and apple (for QuickDraw), for both ProtoVAE and ProtoPNet models.

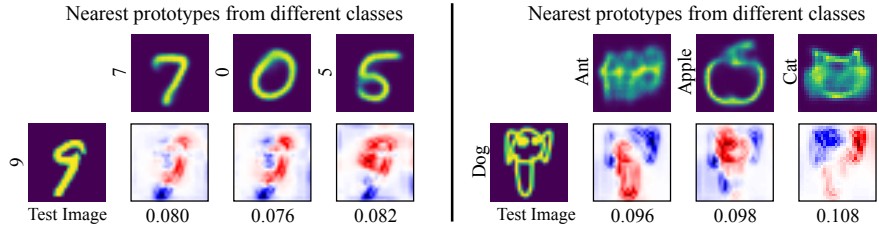

Figure 5: Maximally activated prototypes from three random classes, along with the prototypical explanations for MNIST (left) and QuickDraw (right) datasets.

We also show the closest prototypes from 3 different random classes and their corresponding explainability maps to demonstrate the behavior of explanations for different class prototypes in Fig. 5. Interestingly, the 'dog' image from the QuickDraw dataset resembles an 'ant' prototype for the legs, an 'apple' prototype for the face and a 'cat' prototype for the ears. This information provided by the local explainability maps thus aligns well with human-understandable concepts.

To compare the efficacy of the mapping to the input space learned by our decoder, to methodologies with training-data projection of prototypes [9, 14], we show prototypes along with the 3 closest training images for different datasets in Fig. 6. The prototypes are observed to be the representative of a subset of the respective class. For example, the prototype shown for class '4' of MNIST is representing the subset of '4' with an extended bar, while the 'banana' prototype represents the left facing 'banana' subset, and the 'dog' prototype represents the subset of white dogs on a darker background.

Finally, in order to demonstrate the scalability of ProtoVAE and its applicability on complex higher-resolution real world datasets, we provide an analysis on the CelebA dataset [31] in Sec. S6.10. Note that the less important and fairly diverse features (such as background) appear blurry, while the more important features (skin color, hair color, hair style or age) are crisp and clearly visible.

**Quantitative evaluation**   To quantify the *trustworthiness* of the explanations provided by the proposed model, we calculate the Average Drop (AD) and Average Increase (AI) with respect to local explanation maps and similarity scores for all prototypes [32, 2]. The AD measures the decrease in similarity scores with respect to each prototype when the 50% least important pixels are removed from the images, while AI estimates the ratio of increasing similarity scores. A low AD and a high

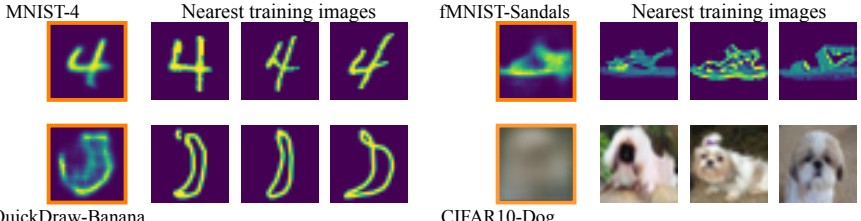

Figure 6: The three closest training images to the learned prototypes for MNIST (class '4'), Quick-Draw (class 'Banana'), fMNIST (class 'Sandals') and CIFAR-10 (class 'Dog'), proving our prototypes representing a "real mean" of subset of classes.

Table 3: AD and AI for quantitative evaluation of explanations of ProtoVAE and ProtoPNet. The reported numbers are means and standard deviations over 5 runs. Best and statistically non-significantly different results are marked in bold.

|  | MNIST | | fMNIST | | CIFAR-10 | | QuickDraw | | SVHN | |
|  | AD | AI | AD | AI | AD | AI | AD | AI | AD | AI |
|---|---|---|---|---|---|---|---|---|---|---|
| ProtoPNet | 3.4±0.3 | **0.6±0.0** | 7.2±0.4 | 0.5±0.0 | 11.6±0.2 | 0.5±0.0 | 2.6±0.1 | 0.7±0.0 | **5.4±0.0** | **0.7±0.0** |
| ProtoVAE | **0.4±0.0** | **0.6±0.0** | **5.1±0.0** | **0.8±0.0** | **6.6±0.0** | **0.7±0.0** | **0.1±0.0** | **0.9±0.0** | 6.1±0.1 | **0.7±0.0** |

AI suggest better performance. These scores are computed as follows:

$$\text{AD} = \frac{100}{NKM} \sum_{i=1}^{N} \sum_{k=1}^{K} \sum_{j=1}^{M} \frac{\max\left(0, \boldsymbol{s}_i(k,j) - \boldsymbol{s}_i^{50\%}(k,j)\right)}{\boldsymbol{s}_i(k,j)}, \ \text{AI} = \sum_{i=1}^{N} \sum_{k=1}^{K} \sum_{j=1}^{M} \frac{[[\boldsymbol{s}_i(k,j) < \boldsymbol{s}_i^{50\%}(k,j)]]}{NKM},$$

where $s_i(k,j)$ is the similarity score of an image $i$ with prototype $j$ of class $k$ (see Eq.1) and $\boldsymbol{s}_i^{50\%}(k,j)$ is the similarity score after masking the 50% least activated pixels according to the prototypical explanation map of prototype $j$. Also, $[[\cdot]]$ are the Iverson brackets which take the value 1 if the statement they contain is satisfied and 0 otherwise.

We report the mean and standard deviation for AD and AI computed over 5 random subsets of 1000 test images for ProtoVAE and ProtoPNet in Table 3. For the grayscale datasets, MNIST and fMNIST, the masked pixels are replaced by 0. For CIFAR-10 and SVHN, they are replaced by random uniformly sampled values. ProtoVAE achieves considerably lower AD and higher or comparable AI for all the datasets. For SVHN, ProtoPNet performs well which we believe is due to the abundance of representative patches in the dataset, thereby improving its explanations.

Finally, we perform a relevance ordering test [33, 12], where we start from a random image and monitor the predicted class probabilities while gradually adding a percentage of the most relevant pixels to the random image according to the local explanation maps. We take 100 random test images and report the average results of change in predicted class probability for all the prototypes in the model. The rate distortion graphs are shown in Fig. 7 for MNIST, QuickDraw and SVHN. We also include two baselines, Random-ProtoPNet and Random-ProtoVAE, where the pixel relevances are ordered randomly. Larger area under the curve indicates better performance. As shown, ProtoVAE's local explanations are able to capture more relevant information than ProtoPNet for all three datasets. Further, for MNIST, ProtoPNet is performing even worse than Random-ProtoPNet, highlighting the lack of trustworthiness in ProtoPNet's explanations.

## 6 Conclusion and Discussion

In this work, we define three properties that act as prerequisites for efficient development of SEMs, namely, *transparency*, *diversity*, and *trustworthiness*. We then introduce ProtoVAE, a prototypical self-explainable method, based on a variational auto-encoder backbone, which addresses these three properties. ProtoVAE incorporates a transparent model and enforces diversity and trustworthiness through the loss functions. In addition to providing faithful explanations, ProtoVAE is able to achieve better predictive performance than its counterpart black-box models.

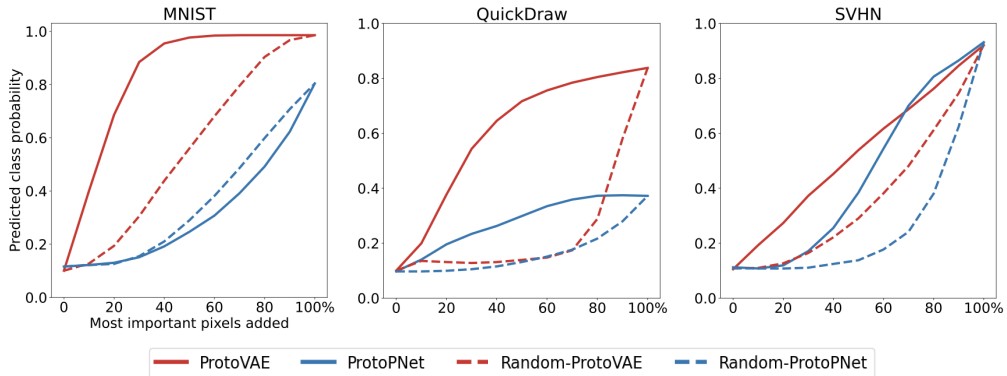

Figure 7: Relevance ordering test for ProtoPNet and ProtoVAE, along with the respective random baselines (Random-ProtoPNet and Random-ProtoVAE). Higher curve suggests better performance of ProtoVAE for all 3 datasets of MNIST, QuickDraw and SVHN.

The main limitation of ProtoVAE is the fixed number of prototypes. This means that the model has to grasp simple as well as more complex classes with the same number of prototypes. For example, in MNIST, there are more variations to be captured by the prototypes in the class '4' than in class '1'. A simple but effective solution is a distance-based pruning procedure, which will be explored in future works. Another approach in sight is to use a prior on the distribution of the prototypical similarities and prioritize some prototypes by controlling the frequency with which each prototype is used in the predictions. Finally, since our global explanations can only be as good as the decoder, one more promising research direction is to leverage more expressive generative models, such as "Very Deep VAEs" [34] and normalizing flows [35] to further improve the scalability of the method to more complex datasets.

## Acknowledgments and Disclosure of Funding

This work was financially supported by the Research Council of Norway (RCN), through its Centre for Research-based Innovation funding scheme (Visual Intelligence, grant no. 309439), and Consortium Partners. The work was further partially funded by RCN FRIPRO grant no. 315029 and RCN IKTPLUSS grant no. 303514. Moreover, the work was partly supported by the German Ministry for Education and Research through the third-party funding project Explaining 4.0 (ref. 01IS20055).

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
