# OpenReview forum: "ProtoVAE: A Trustworthy Self-Explainable Prototypical Variational Model"
_NeurIPS.cc/2022/Conference — NeurIPS 2022 Accept_

### Official Review · Reviewer_zZon · 2022-07-05

**Rating:** 6
**Confidence:** 4
**Soundness:** 3 good
**Presentation:** 3 good
**Contribution:** 2 fair

**Summary:**

The authors introduce ProtoVAE, a VAE-based framework for interpretable classification. Specifically, the authors propose to learn class-specific prototypes in the VAE's latent space. For a given sample, the distance to those prototypes is then used by a linear layer to classify that sample. Since the authors additionally train a decoder, the learnt prototypes can be visualised in input space, which makes the prototypes and thus the classification pipeline easily interpretable.

To evaluate their approach, the authors compare to various other self-explainable models with a focus on ProtoPNet and discuss advantages w.r.t. classification accuracy and interpretability of their proposed approach.

**Questions:**

Please see the list of weaknesses, I would appreciate if the authors could address my concerns.

Specifically:
- Can we hope to scale this approach to larger datasets? How would this approach compare to ProtoPNet on, e.g., CUB?
- How does the interpretability of ProtoVAE compare to ProtoPNet if the prototypes cannot be visualised anymore (e.g., prototypes are already difficult to interpret on CIFAR10)?
- What do we learn from the LRP-based explanations? The heatmaps do not seem to be easily interpretable and even somewhat conflicting when it comes to highlighting the similarity to the different prototypes (Fig. 4, top row).
- The authors say that the blurriness of the CIFAR prototypes can be reduced by a larger number of prototypes. Is there experimental evidence for this? What is the impact of the L2 reconstruction loss on this?

**Limitations:**

To me, there are no obvious negative societal impacts that the authors would need to address in more detail. However, as expressed in the previous points, I believe the manuscript would benefit significantly from a more detailed discussion of the limitations w.r.t. scalability and current bottlenecks.

**Strengths And Weaknesses:**

## Strengths
- The proposed approach is conceptually simple, but nevertheless yields better accuracy than other self-explainable models on the evaluated datasets and allows for a direct visualisation of the used prototypes in input space.
- By using a VAE, the proposed approach yields an easily interpretable latent space and the model can interpolate between different input samples.
- Further, the model allows for both global explanations (the learnt prototypes) as well as local explanations (distances to learnt prototypes of a given sample).
- The manuscript is generally well-written and easy to read.


## Weaknesses

- ProtoPNet was developed for a more complex dataset (larger images, fine-grained classification). In contrast, the proposed approach evaluates only on much simpler datasets — a discussion on potential scalability issues is missing.
- Continuing the previous point: the interpretability of the proposed model crucially depends on the quality of the decoder. However, on CIFAR10 the prototypes already become much less interpretable (Fig. 6, dog prototype or Fig. 2 CIFAR10 prototypes). Can we even hope for such an approach to be scaled to 'relevant' datasets in terms of interpretability?
- Apart from the interpretability, the accuracy on CIFAR10 also seems very limited. I would appreciate if the authors could comment on the major limiting factors w.r.t. classification performance. A minor point related to this: the comparison to the black-box encoders is confusing (Table 2). While I understand that the authors introduce this as a relevant comparison in Definition 3 (i), this comparison seems to implicitly suggest that ProtoVAEs outperform black-box models. However, 84.6% accuracy on CIFAR10 is, of course, far from what DNNs can achieve.
- A core feature of the ProtoPNet is to learn and compare to *local* prototypes ("This looks like that"). In contrast, the ProtoVAE is designed to learn full-image prototypes, which limits its flexibility. I would appreciate if the authors could comment on learning local prototypes, especially given that the comparison to ProtoPNet is so central in the manuscript.
- While I understand the motivation for the local explanations according using LRP, I am unsure about their added benefit w.r.t. the interpretability of the model decision. I would appreciate if the authors could discuss this in more detail, specifically because it seems difficult to parse those explanations: what do we learn e.g., from the heatmaps in Fig. 4 (top row)? (minor: and what is the colour coding exactly? red positive, blue negative?) Further, LRP is a general framework that encompasses many potential rules for backpropagating relevance scores. It is unclear from the text which rules were chosen.


Minor:
- There are other generative classification approaches, maybe most notably normalizing flows (e.g., Mackowiak et al. 2021), that might be relevant for a more complete contextualisation.

---

> ### Author Response · Authors · 2022-08-02
> **Response to Reviewer zZon Part 3**
>
> ### Blurriness of CIFAR prototypes
>
> We would like to thank the reviewer for motivating us to go beyond our intuition and conduct an extensive study on all the datasets which finally confirm our statement: Adding more prototypes does improve the quality of the global explanations as well as it favors diversity within the classes. Additionally, increasing the weight of the reconstruction loss does help.
> For example, in Figure 10 of the revised supplementary material, we can see the prototypes for the grape class of QuickDraw refine and take shape as their number increases.
> Despite the poor quality of CIFAR-10's reconstructions, a similar phenomenon is nonetheless observable (Figure 11). When trained with three prototypes per class, some prototypes are degenerate, what does not happen when their number increases. If we focus on the automobile class, it is not apparent from the prototypes that a car is learned when only considering three learned prototypes. However, when their number increases, the car becomes more evident and different orientations of the car are captured by each prototype.
>
> The impact of the L2 reconstruction is analyzed in Figures 12, 13 and 14 in the supplementary material. We show the reconstructed prototypes for QuickDraw obtained with 4 different models trained with the weights of 0.1, 1, 10 and 100 for L2 loss. Similarly, we show the prototypes for CIFAR-10 for weights 0.1 and 10. For both the datasets, we demonstrate that the prototype reconstructions tend to be sharper with higher weight of the L2 reconstruction.
>
> Due to length constraints and the extent of these studies, we have added them discussion to the supplementary material in Section S6.8 and S6.9, respectively.
>
> ### More advanced generative models
> As mentioned in the "Scalability" part of this response, we believe that the use of more advanced generative models as backbone is a promising direction to explore in future work in order to ensure better reconstruction of prototypes for more complex datasets. We have expanded the limitation part in the revised manuscript (Section 6) to highlight this further.

---

> > ### Comment · Reviewer_zZon · 2022-08-09
> > **Post-rebuttal response**
> >
> > I thank the authors for the detailed response and the work put into updating the manuscript and explicitly addressing my concerns. I believe that the modifications significantly improved the manuscript and the authors were able to clarify most of my questions. Given this, I increase my score to weak accept — however, I share the view of reviewer 8Fru that some crucial questions (especially regarding scalability and the interpretability on more complex datasets) remain unanswered, which also gives me reason to still be hesitant.
> >
> > I wish the authors all the best for their submission.

---

> > > ### Author Response · Authors · 2022-08-09
> > > **Second response to Reviewer zZon**
> > >
> > > Thank you again for your encouraging feedback and your suggestions. We are happy to hear that we addressed your concerns and hope that we in the following can further address your concern regarding the applicability of our method to real-world complex tasks.
> > >
> > > We do understand the concern that the reviewers raise with regards to the real-world applicability of our method and have therefore included additional results on the CelebA dataset in the paper. CelebA is another common real-world image generation dataset consisting of celebrity images, which has a considerably higher resolution (178×218) than any of the previously included datasets. The results and a discussion are included in Section S6.10 in the supplementary material, highlighting the quality of the prototypes that our model learns and showcasing its applicability to higher-resolution real-world datasets.
> > >
> > > We have in the past week further tried to get similar high-quality and crisp results for CIFAR-10, but unfortunately, CIFAR-10 can be considered quite a difficult dataset due to its high variation and low-resolution, which quickly leads to blurry image generation results. While recent generative approaches have shown promising performance on CIFAR-10, an extensive exploration of these backbones for our approach is infeasible given their multiple week long training runs (for Very Deep VAEs). We hope that the reviewer will consider the problem of getting crisp CIFAR-10 reconstructions out of the scope of this work, given the newly added results on another high-resolution real-world dataset.
> > >
> > > To further support our claim that the expressiveness of the backbone is crucial for reconstructing sharp prototypes, we have additionally added Section S6.11 in the supplementary material, which shows and discusses the test reconstructions for the MNIST, fMNIST, CIFAR-10, QuickDraw and SVHN datasets.
> > >
> > > We have updated the paper and supplementary material with the new results.

---

> ### Author Response · Authors · 2022-08-02
> **Response to Reviewer zZon Part 2**
>
> ### Classification performance
> Thank you for pointing out the ambiguity of the text which could lead readers to think that ProtoVAE outperforms any black-box model.
> The black-box counterpart refers to the feed-forward classifier consisting of the same encoder as ProtoVAE followed by a linear classifier, i.e., a classical CNN.
> We have now updated the Section 5 (Baselines) of the manuscript to prevent the said ambiguity.
> Note that ProtoVAE makes use of a relatively simple architecture, explaining the relative low performance of 84.6\% for ProtoVAE and of 83.9\% for its black-box CNN counterpart. However, a state-of-the-art classifier on CIFAR-10 would indeed obtain results of up to 99\% accuracy.
> In Section 6.2 of the supplementary material, we experimented with a ResNet-18 as an encoder and achieve 80.5\% accuracy. Albeit low, we would like to state that we used simple augmentation strategies, as mentioned in the supplementary material (Section S3), and do not employ learning rate scheduler, weight decay or momentum. The state-of-the art results achieved on CIFAR-10 employ complex training strategies which highly impact the achieved accuracy [1].
> The main objective of this work is to demonstrate the effectiveness of the proposed method to learn a trustworthy and transparent probabilistic prototypical space. Therefore, we did not aim to reach state-of-the-art classification performance but to compare accuracies with black-box counterparts as well as to compare to other self-explainable baselines while sharing when possible the same training strategies.
>
> [1] Moreau et al. "Benchopt: Reproducible, efficient and collaborative optimization benchmarks", arXiv, 2022.
>
> ### Local explanations vs full-image prototypes
> Although local prototypes seem more intuitive, they come with their own set of challenges. In our experience and as shown in Figure 7 in the supplementary material, the patches learned by ProtoPNet as prototypes are difficult to interpret on their own and full prototype images are required for achieving meaningful interpretations. Further, patch prototypes learned by ProtoPNet have been questioned in the literature to be spatially imprecise due to the large receptive field of the backbone architecture [1]. On the other hand, ProtoVAE allows more flexibility by allowing prototypes to be fully learnable and provides more trustworthy interpretations with the help of the learnt decoder.
>
> [1] Gautam et al. "This looks more like that: Enhancing self-explaining models by prototypical relevance propagation" arXiv, 2021.
>
> ### Interpretability of local LRP explanation maps
> Thank you for pointing out this gap in the discussion about the local explanations. The motivation for computing local explanation maps is to see which parts of an image are activated by different prototypes. The red pixels correspond to positive relevance and the blue pixels correspond to negative relevance. In Figure 4 top row for the test image of class '5', the first prototype has a similar top bar as the test image (straight horizontal while for the other prototypes, it is curved or tilted) and therefore has a positive relevance there and negative relevance at the curved area due to high dissimilarity in that region. In Figure 5 (left), three prototypes from different classes and their local explanation maps are shown. As observed, the test image of class '9' has a similar slant line as digit '7', some similar curved areas as the digit '0' and a similar top area as the digit '5'. Further in Figure 7 (right), the dog test image has similar legs to those of an ant, face shape to that of an apple and ear to that of a cat.
> We have used the LRP composite rule, which is known as best practice [1] to compute local explanation maps. Following this, the LRP\_{\alpha\beta} rule is applied to the convolutional layers and the Deep Taylor Decomposition based rule DTD\_{z^B} [2] is applied to the input features. We have correspondingly updated the manuscript to clarify this further in Section 4.3.
>
>
> [1] Kohlbrenner, et al. "Towards best practice in explaining neural network decisions with lrp". IJCNN 2020.
>
> [2] Kauffmann et al. "Towards explaining anomalies: A deep taylor decomposition of one-class models". Pattern Recognition, 2020.

---

> ### Author Response · Authors · 2022-08-02
> **Response to Reviewer zZon Part 1**
>
> Thank you for the detailed feedback. We are glad that you appreciate our proposed method and acknowledge its ability to achieve better accuracy while being conceptually simple. In the following, we happily address your concerns:
>
> * **Scalability - Predictive performance**: A discussion on potential scalability issues is missing. Can this approach scale to larger datasets?
> * **Scalability - Interpretability**: Interpretability of the proposed model crucially depends on the quality of the decoder.
> * **Classification performance**: Apart from the interpretability, the accuracy on CIFAR-10 also seems very limited. Comment on the major limiting factors w.r.t. classification performance.
> * **Local explanations vs full-image prototypes**: I would appreciate if the authors could comment on learning local prototypes, especially given that the comparison to ProtoPNet is so central in the manuscript.
> * **Interpretability of local LRP explanation maps**: Added benefit of LRP w.r.t. the interpretability of the model decision. Further, it is unclear which LRP rules were chosen.
> * **Blurriness of CIFAR prototypes**: Show that less blurry prototypes can be obtained if one extracts more prototypes to be extracted per class. What is the impact of the L2 reconstruction loss on this?
> * **More advanced generative models**: Other generative classification approaches (e.g., Mackowiak et al. 2021) might be relevant for a more complete contextualisation.
>
>
> ### Scalability - Predictive performance
> In terms of predictive performance, ProtoVAE consistently outperforms ProtoPNet on various datasets with different degree of complexity undoubtedly thanks to the fully learnable and diverse prototypes.
> The latter gives our model flexibility to organize the latent space to ensure both good predictions and the transparency thereof, as opposed to ProtoPNet where prototypes are forced to represent fixed training images.
> We would thus expect ProtoVAE to perform on par or better than ProtoPNet on the CUB-200 dataset from a predictive performance perspective.
>
> ### Scalability - Interpretability
> In our work, we chose to leverage a VAE backbone due to its simplicity and training stability to demonstrate how diverse and differentiable prototypes can be learned in a transparent probabilistic space to explain a classifier. However, you are indeed correct that our simple VAE backbone will struggle on more complex high-resolution datasets such as CUB-200. Generation/reconstruction of high-quality complex images is a highly active field of research and more advanced backbones could be adapted to ensure good reconstruction also for more complex datasets. These can be either directly based on VAEs, such as the "Very Deep VAEs" [1], which can produce high-quality high-resolution reconstructions, or could leverage closely related methods such Normalizing Flows [2], Inverse Autoregressive Flows [3] or Probabilistic Deep Image Prior [4]. We have added a discussion of this to the limitations in Section 6 of the revised manuscript.
>
> Nonetheless, the robust local explanability maps (Figure 4) produced by ProtoVAE offer a way to explain the predictions or the similarity scores in a situation where the visualizations of the prototypes are not very informative. Indeed, the heatmaps highlight the parts of the test image that are important for the similarity score to a given prototype and ultimately for the class prediction.
> Note that such a strategy cannot be pursued with ProtoPNet as its local explanability maps are often uninformative [5].
>
> Finally, a workaround to the poor reconstruction is to sample neighboring training images, as incorporated by FLINT, SENN or ProtoPNet, and infer shared characteristics among them. An example of nearest training images activated by prototypes is shown in Figure 6 in the main text.
>
> ---
>
> [1] Child. "Very deep vaes generalize autoregressive models and can outperform them on images." ICLR 2021.
>
> [2] Rezende and Mohamed. "Variational inference with normalizing flows." ICML 2015.
>
> [3] Kingma, et al.. "Improved variational inference with inverse autoregressive flow". NeurIPS 2016.
>
> [4] Cheng et al. "A Bayesian Perspective on the Deep Image Prior", CVPR 2019.
>
> [5] Gautam et al. "This looks more like that: Enhancing self-explaining models by prototypical relevance propagation" arXiv, 2021.

---

### Official Review · Reviewer_8Fru · 2022-07-10

**Rating:** 6
**Confidence:** 4
**Soundness:** 3 good
**Presentation:** 4 excellent
**Contribution:** 3 good

**Summary:**

This paper introduces ProtoVAE, a novel self-explainable architecture which learns meaningful and diverse prototypical explanations via an architecture that exploits representations learned by a mixture of VAEs. ProtoVAE is able to maintain competitive performance, even against black-box models, while learning to produce prototypical explanations outside the training set in an end-to-end fashion. This allows ProtoVAE to produce clearer local explanations than competing approaches (via attribution maps for each prototype) as well as useful prototypical images from which an explanation can be built. This method is evaluated in a diverse evaluation set of five tasks and compared against existing state-of-the-art methods such as ProtoPNet.

**Questions:**

After reading the paper a few times, I would appreciate it if the authors could clarify the following:

1. My biggest concern with this work is that it seems that it does not generalize well in complex tasks such as CIFAR-10 (with prototypes being very blurry). Even in some simpler tasks like QuickDraw, we see that prototypes within a class are able to capture the main features of that class but any prototype-specific details remain extremely blurry (e.g., see nearest prototypes in the upper right quadrant of Figure 4). Nevertheless, line 230 indicates that less blurry prototypes can be obtained if one extracts more prototypes to be extracted per class. Is that something that can be shown in the main body of the paper? If so, I believe it would make your evaluation much stronger and qualm any doubts about the applicability of this method to non-synthetic tasks.
2. Is it possible to include more modern methods than ProtoPNet (e.g., SITE) in the comparison during evaluation? I understand that this may entail some further experimentation to be done but it is unusual to have such empty entries in an evaluation table (as in Table 2) because the papers discussing these works did not include the same tasks as the ones used in this paper. Furthermore, it is important to get an understanding of how this method compares to other prototypical SENs that also learn sample-level prototypes (rather than patch prototypes as ProtoPNet does). This is not a blocking request as the paper does a good job at evaluating ProtoVAE against ProtoPNet but it would again make your evaluation stronger.

In terms of suggestions for the presentation, the paper is extremely well-written and clear. Nevertheless, the following typos may be worth addressing for a final version:

1. Line 101: “… takes as input features of several hidden layers” should probably be “… takes as input features several hidden layers”.
2. Line 121: “embedding” should be “embeddings”.
3. Line 130: $g(z_i)$ should be $g(\mathbf{z}_i)$ for notational consistency.
4. In several parts of the paper and the supplementary material, “supplementary” is used instead of “supplementary material”.

**Limitations:**

The paper does a good job at discussing some of the crucial limitations of their method. For example, it discusses how some tasks result in blurry prototypes in the experimentation section and briefly elaborates on the limitations of fixing the number of prototypes per class in the conclusion. The paper also discusses possible negative societal impacts in its supplementary material.

**Strengths And Weaknesses:**

Thank you for the very insightful and interesting work presented in this paper. After carefully reading the paper, I believe that its main strengths are:

1. It tackles an important gap in prototypical XAI methods (that of designing a transparent, diverse, and trustworthy end-to-end prototypical model) and does a good job at motivating how this is reached through careful definitions for the model’s desiderata.
2. The evaluation of the method is broad. It evaluates the proposed method both quantitative to qualitative across tasks with a varying degree of difficulty.
3. The paper is extremely well-written and the presentation is clear.
4. The introduction of a VAE-like architecture for learning out-of-training-set prototypes is novel and potentially interesting for future work to further explore.

In contrast, I believe the following elements are weaknesses of the presented work:

1. Against common practice, some results do not have error bars in them (e.g., SITE results in Table 2 and all results in Table 3).
2. In SVHN, evaluation results for most baselines are missing. Similarly, when evaluating ProtoVAE against SITE (a potentially important contender for the method proposed), evaluation results are missing for 3 of the selected tasks.
3. The evaluation of the method in real-world complex tasks shows that its prototypes are hard to interpret: prototypes learned in CIFAR-10 (as seen in Figure 2) seem to be extremely blurry and hard to interpret (as they seem to be some sort of mean of samples in that class). This casts some doubt on the applicability of this method to real-world distributions.

---

> ### Author Response · Authors · 2022-08-02
> **Response to Reviewer 8Fru Part 3**
>
>
> ### Less blurry prototypes as number increases
> We would like to thank the reviewer for motivating us to go beyond our intuition and conduct an extensive study which confirms our statement: Adding more prototypes does improve the quality of the global explanations as well as it favors diversity within the classes.
> For example, in Figure 10 of the revised supplementary material, the prototypes for the grape class of QuickDraw become more precise and take shape as the number of prototypes per class increases.
> Despite the poor quality of CIFAR-10's reconstructions, a similar phenomenon is nonetheless observable (Figure 11) when the prototypes per class are increased from 3 to 20.
> With only 3 prototypes per class, prototypes are degenerate and it is hard to guess that automobile prototypes depict cars.
> However for 20 prototypes per class, cars becomes more recognizable and their orientation becomes more perceivable.
>
> We hope that this study further strengthens the evaluation and addresses your doubts.
> Due to length constraint, we have added this study to the supplementary material in Section S6.8.

---

> > ### Comment · Reviewer_8Fru · 2022-08-06
> > **Reply to rebuttal comments and updated submission**
> >
> > Dear Authors,
> >
> > I would like to begin by thanking you for taking the time to read all of our reviews and address some of our concerns through your rebuttal and the updated paper. As mentioned in my original review, I find this work exciting and think there is potential in this submission. I would also like to acknowledge and thank you for addressing my concerns regarding (i) missing error bars, (ii) evaluation baselines, and (iii) some of the questions I had regarding the applicability and blurriness of some prototypes. These are all great improvements to the manuscript. After going over the new submission and your replies, I can't help it but still be a bit hesitant about the applicability of this method in real-world settings. More specifically, I am still further concerned about the following items:
> >
> > - Thank you for your new experiments on the effect that the number of prototypes has on their blurriness. While it is clear, as you rightly point out, that the more prototypes per class the crisper these results are (with the weird artifacts seen in the "Cat" and "Airplane" class when using 3 prototypes being an extreme example), it is unfortunately still hard to intuitively make sense as to what each specific prototype represents even when using 20 prototypes per class. This, unfortunately, is a relatively big limitation as the purpose of this method is for it to be able to generate meaningful explanations for practitioners. It therefore remains to be seen whether CIFAR-10 is a corner-case real-world example that happens to work badly with this method or if it is the case that the proposed model generally does not generate clear prototypes for non-synthetic/complex distributions.
> > - Similarly, thank you for your new results in Section S6.7 of the appendix. These results are certainly interesting however I still fail to fully see how such maps can be useful in practice when the prototypes themselves are hard to interpret. For example, the bottom right image of Fig 9 shows an automobile prototype and its corresponding LRP map, however it is not very clear what it is highlighting and how this would therefore constitute an actionable/insightful explanation in practice. In particular, it is unclear how such an explanation would clarify what the prototype represents (as, for example, seeing that the prototype for an airplane focuses on the actual airplane in the image is useful but does not tell us much more about the prototype itself). Similar comments apply to the image in the top right and its corresponding "airplane" prototype.
> > - If it is true that better encoding/decoding models (e.g., "Very Deep VAEs") could lead to better, more crisp prototypes, then this would absolutely put my concern to rest. Nevertheless, the current evidence as presented in this paper does not support this and it therefore remains an open question to determine whether this is actually true. I understand running such experiments in the rebuttal window would've been infeasible, and therefore I did not expect them in the revised version. Nevertheless, such results would help build a much stronger case for this method's applicability even if they can be shown only for CIFAR-10 as a representative of a real-world task.
> > - It is worth pointing out that although the updated manuscript is already exceeding the page limit, it would make the work done for this manuscript more cohesive if the different appendices that were added during rebuttals would be at least mentioned (with a very brief description of their purpose/main findings) in the main text of the paper. Otherwise, the paper depends on the reader's curiosity to understand some of the interesting and important results discussed in the Appendix.
> >
> > Because of all of these reasons, I am leaning towards maintaining my current "weak acceptance" score as is. This is because, as this paper currently stands, it presents a very interesting and exciting idea (presented in a clear manner) but the lack of evidence for real-world task applicability of its main method limits the impact of this work. Nevertheless, I am happy to be convinced otherwise by you or other reviewers/ACs that this on its own is a strong enough contribution to merit a score increase to an "accept" if I may have misunderstood something or missed something crucial.
> >
> > I wish you the best of luck with this submission!

---

> > > ### Author Response · Authors · 2022-08-09
> > > **Second response to Reviewer 8Fru**
> > >
> > > Thank you again for your constructive suggestions and we are glad that we have addressed most of your concerns. In the following, we further respond to your concern regarding real-world applicability.
> > >
> > > We do understand the concern that the reviewers raise with regards to the real-world applicability of our method and have therefore included additional results on the CelebA dataset in the supplementary material. CelebA is another common real-world image generation dataset consisting of celebrity images, which has a considerably higher resolution (178×218) than any of the previously included datasets. The results and a discussion are included in Section S6.10 in the supplementary material, highlighting the quality of the prototypes that our model learns and showcasing its applicability to higher-resolution real-world datasets.
> > >
> > > We have in the past week further tried to get similar high-quality and crisp results for CIFAR-10, but unfortunately, CIFAR-10 can be considered quite a difficult dataset due to its high variation and low-resolution, which quickly leads to blurry image generation results. While recent generative approaches have shown promising performance on CIFAR-10, an extensive exploration of these backbones for our approach is infeasible given their multiple week long training runs (for Very Deep VAEs). We hope that the reviewer will consider the problem of getting crisp CIFAR-10 reconstructions out of the scope of this work, given the newly added results on another high-resolution real-world dataset.
> > >
> > > To further support our claim that the expressiveness of the backbone is crucial for reconstructing sharp prototypes, we have additionally added Section S6.11 in the supplementary material, which shows and discusses the test reconstructions for the MNIST, fMNIST, CIFAR-10, QuickDraw and SVHN datasets.
> > >
> > > We have updated the supplementary material with the new results and have also added mentions of all the supplementary results to the paper. While the revised manuscript is slightly exceeding the original 9 page limit, an additional page is available for accepted papers in order to address the reviewers concerns. In such a case, we will be well within the page limit.

---

> ### Author Response · Authors · 2022-08-02
> **Response to Reviewer 8Fru Part 2**
>
> ### Applicability on real-world complex tasks
> In our work, we chose to leverage a VAE backbone due to its simplicity and training stability to demonstrate how diverse and differentiable prototypes can be learned in a transparent probabilistic space to explain a classifier. However,  our simple VAE backbone will struggle on more complex high-resolution datasets. Generation/reconstruction of high-quality complex images is a highly active field of research and more advanced backbones could be adapted to ensure good reconstruction also for more complex datasets. These can be either directly based on VAEs, such as the "Very Deep VAEs" [1], which can produce high-quality high-resolution reconstructions, or could leverage closely related methods such Normalizing Flows [2], Inverse Autoregressive Flows [3] or Probabilistic Deep Image Prior [4]. We have added a discussion of this to the limitations in Section 6 of the revised manuscript.
>
> Nonetheless, the robust local explanability maps (Figure 4) produced by ProtoVAE offer a way to explain the predictions or the similarity scores in a situation where the visualizations of the prototypes are not very informative. Indeed, the heatmaps highlight the parts of the test image that are important for the similarity score to a given prototype and ultimately for the class prediction.
> We added to the supplementary material (Section S6.7 and Figure 9) an example similar to Figure 4 in the main paper but based on CIFAR-10 dataset. For each test image, the LRP maps for 2 prototypes from different classes are shown to analyse the area of interest in the test image by different class prototypes. Interestingly, the 'horse' class prototype can be seen activating a horse-shaped shadow on an 'airplane' test image and an 'airplane' prototype can be seen activating wheel-like features in a 'ship' test image.
> Note that such a strategy cannot be pursued with ProtoPNet as its local explanability maps are often uninformative [5].
>
> Finally, a workaround to the poor reconstruction is to sample neighboring training images, as incorporated by FLINT, SENN or ProtoPNet, and infer shared characteristics among them. An example of nearest training images activated by prototypes is shown in Figure 6 in the main text.
>
>
> ---
>
> [1] Child. "Very deep vaes generalize autoregressive models and can outperform them on images." ICLR 2021.
>
> [2] Rezende and Mohamed. "Variational inference with normalizing flows." ICML 2015.
>
> [3] Kingma, et al.. "Improved variational inference with inverse autoregressive flow". NeurIPS 2016.
>
> [4] Cheng et al. "A Bayesian Perspective on the Deep Image Prior", CVPR 2019.
>
> [5] Srishti Gautam, Marina M. C. Ho ̈hne, Stine Hansen, Robert Jenssen, and Michael Kampffmeyer, “This looks more like that: Enhancing self-explaining models by prototypical relevance propagation,” arXiv, 2021.
>
> ### Prototype interpretation
> First of all, it is important to accept that images of reconstructed prototypes will always contain blurry areas as these are regions with a lot of variations within the neighborhood around the prototypes.
> Despite crisp reconstructions of test and training images, the last prototype of ProtoVAE for the 7s of MNIST (Figure 2 bottom right) has a blurry crossing bar suggesting that its length can vary.
> The case of CIFAR-10 is particular as the model fails to faithfully reconstruct even training images.
> As for the apples in QuickDraw (Figure 4 top right), the prototypes show only few variations because the shape of an apple is quite consistent.
> Notice that the stem, which can vary in length, orientation and may or may not have leaves, is always blurry.
>
> In Figure 2, we show how ProtoVAE learns diverse prototypes between and within the classes.
> The case of the sandals in FMNIST (Figure 2, second plot, second line) is especially interesting.
> Although none of the prototypes is very sharp, different subgroups are distinguishable: flat sandals (right), high heels with simple (left) or complex (center) closing.
> As for the automobile class of CIFAR-10 (Figure 2, fourth plot, first line), one can recognize different orientations and colors.

---

> ### Author Response · Authors · 2022-08-02
> **Response to Reviewer 8Fru Part 1**
>
> Thank you for your careful revision and for acknowledging the importance of our work in filling an important gap in prototypical XAI methods, as well as appreciating the thorough evaluation and novelty of the proposed method in terms of learning out-of-training-set prototypes. In the following, we respond to your questions:
>
> * **Missing error bars in Table 2 and 3, and baseline results on SVHN**: Some results do not have error bars in them. Evaluation results for most baselines are missing for SVHN. SITE evaluation results are missing for some tasks.
> * **Modern comparisons**:  Comparison to other prototypical SENs that also learn sample-level prototypes.
> * **Applicability on real-world complex tasks**: Prototypes are hard to interpret for more complex tasks.
> * **Prototype interpretation**: Prototypes within a class are able to capture the main features of that class but any prototype-specific details remain extremely blurry.
> * **Less blurry prototypes as number increases**: Show that less blurry prototypes can be obtained if one extracts more prototypes to be extracted per class.
>
> In addition, we have corrected the typos in the updated manuscript accordingly.
>
>
> ### Missing error bars in Table 2 and 3, and baseline results on SVHN
> As these two concerns/questions are related, we address them jointly in the following. In the original manuscript, we decided to report the original results for FLINT, SENN and SITE. This was to the benefit of the baselines, since each baseline uses different architectures and training schemes likely optimized for the model. There were therefore missing values for SVHN since it was not used in any of the original papers. However, we understand that this can be viewed as unusual and have therefore revised Table 2 to include accuracies using our architecture on all tasks (including SVHN) and for all methods but SITE. The code for the latter has not been released and we have so-far been unable to obtain it directly from the authors.
> These new results confirm that ProtoVAE is also more accurate on SVHN than the more modern baselines FLINT and SENN.
>
> In the revised manuscript, all the entries of Table 2 are means over four runs with standard deviation, except for SITE.
> Accordingly, we have also added error bars for all results in Table 3, where we report the mean and standard deviation for AD and AI computed over 5 random subsets of 1000 test images for ProtoVAE and ProtoPNet.
> Finally, we have transferred the previous version of Table 2, to the supplementary material.
> For convenience we have included the updated Table 2 below:
>
>
> **Table 2**
> Results for accuracy (in \%) for ProtoVAE and comparison with other state-of-the-art methods. The reported numbers are means and standard deviations over 4 runs. Best and statistically non-significantly different results are marked in bold. *Results for SITE are taken from the original paper and thus based on more complex architectures.
>
> |           | Black-box       |    FLINT        | SENN        | SITE*   | ProtoPNet   | ProtoVAE        |
> |-----------|-----------------|-----------------|-------------|---------|-------------|-----------------|
> | MNIST     | 99.2 ± 0.1      | __99.4 ± 0.1__  | 98.8 ± 0.7  | 98.8    | 94.7 ± 0.6  | __99.4 ± 0.1__  |
> | fMNIST    | 91.5 ± 0.2      | 91.5 ± 0.2      | 88.3 ± 0.3  | -       | 85.4 ± 0.6  | __91.9 ± 0.2__  |
> | CIFAR-10  | 83.9 ± 0.1      | 79.6 ± 0.6      | 76.3 ± 0.2  | 84.0    | 67.8 ± 0.9  | __84.6 ± 0.1__  |
> | QuickDraw | 86.7 ± 0.4      | 82.6 ± 1.4      | 79.3 ± 0.3  | -       | 58.7 ± 0.0  | __87.5 ± 0.1__  |
> | SVHN      | __92.3 ± 0.3__  | 90.8 ± 0.4      | 91.5 ± 0.4  | -       | 88.6 ± 0.3  | __92.2 ± 0.3__  |
>
> ### Modern comparisons
> Thank you for the suggestion. We have in the revised version improved the comparison to the modern prototypical SENs (FLINT and SENN) by evaluating their performance additionally on the SVHN dataset (included in Table 2).
> As for comparing global explanations, neither FLINT nor SENN are designed to directly produce input space representations of their prototypes. These are limited to selecting the nearest training example.

---

### Official Review · Reviewer_iezD · 2022-07-19

**Rating:** 6
**Confidence:** 3
**Soundness:** 2 fair
**Presentation:** 2 fair
**Contribution:** 2 fair

**Summary:**

The authors presented a nice piece of work in this paper towards improving deep model explainability through concept learning with prototypes. The proposed method is designed to generate explanations that are transparent, diverse and trustworthy. The predictive performance of the proposed method is better compared to related methods with various digit as well as natural image datasets.

**Questions:**

Questions: a) VAEs are not well known for reconstruction quality of bigger size images or complex images with more prototypes compared to the 5 datasets used for experimentation in this paper. While this method shows nice results with the said datasets, would this method scale to more complex datasets?

b) In L94, there is a typo at "incorporates generates".

**Ethics Review Area:**

["I don’t know"]

**Limitations:**

The authors have addressed the limitations, but did not comment about any potential negative societal impact of their work.

**Strengths And Weaknesses:**

Strength: a) The proposed self-explainable method is capable of producing explanations that have some desirable properties i.e. transparency, diversity and trustworthiness. The properties are induced in the generated explanations through different loss terms used for model training.

b) The model achieves better accuracy compared to related methods with 5 different datasets. That improves the usability of the proposed method as the explanations can be generated now without sacrificing the predictive performance much.

Weaknesses: a) I think the technique followed here to generate the prototypes should be explained more clearly. Did you simply use the decoder to reconstruct the prototypes?

b) The structure of the prototypes are not clearly understood from fig1 and the number of prototypes as well. The information that every column vector of the matrix (phi_k) are the prototypes is not articulated through fig1.

---

> ### Author Response · Authors · 2022-08-02
> **Response to Reviewer iezD**
>
> We thank the reviewer for the constructive feedback and for acknowledging the quality of explanations as well as the predictive performance achieved by our method. We are delighted to address the concerns and questions raised by the reviewer in the following:
>
> * **Reconstruction of prototypes**: Elaborate the technique followed here to generate the prototypes. Was the decoder used to reconstruct the prototypes?
> * **Structure of Fig 1**: The structure of the prototypes are not clearly understood from fig1 and the number of prototypes as well.
> * **Extension to complex datasets**: VAEs reconstruction quality for bigger size images or complex images with more prototypes.
> * **Negative societal impact**: Missing Negative societal impact.
> In addition, we have corrected the typo in the updated manuscript accordingly.
>
> ### Reconstruction of prototypes
> Thank you for letting us know that this was not clear to the reader. ProtoVAE is designed to have the inherent capability to reconstruct and thus produce visualization of its prototypes via the decoder, which is trained to approximate the input distribution. The operations have been clarified in Section 4.3 of the revised manuscript.
>
> ### Structure of Fig 1
> Thank you for pointing out the missing index of phi\_k in Figure 1 representing the prototypes. It should have been phi\_kj, as in the text. We revised the figure and chose to explicitly represent that there are M prototypes for each of the K classes. In addition, we have also added labels to further explicit some parts of the network and extended the caption. We hope that this new figure gives a better overview of the operations involved in ProtoVAE.
>
> ### Extension to complex datasets
> In our work, we chose to leverage a VAE backbone due to its simplicity and training stability to demonstrate how diverse and differentiable prototypes can be learned in a transparent probabilistic space to explain a classifier. However, you are indeed correct that our simple VAE backbone will struggle on more complex high-resolution datasets. Generation/reconstruction of high-quality complex images is a highly active field of research and more advanced backbones could be adapted to ensure good reconstruction also for more complex datasets. These can be either directly based on VAEs, such as the "Very Deep VAEs" [1], which can produce high-quality high-resolution reconstructions, or could leverage closely related methods such as Normalizing Flows [2], Inverse Autoregressive Flows [3] or Probabilistic Deep Image Prior [4]. We have added a discussion of this to the limitations in Section 6 of the revised manuscript.
>
> ---
>
> [1] Child. "Very deep vaes generalize autoregressive models and can outperform them on images." ICLR 2021.
>
> [2] Rezende and Mohamed. "Variational inference with normalizing flows." ICML 2015.
>
> [3] Kingma, et al. "Improved variational inference with inverse autoregressive flow". NeurIPS 2016.
>
> [4] Cheng et al. "A Bayesian Perspective on the Deep Image Prior", CVPR 2019.
>
> ### Negative societal impact
> The negative societal impact discussion is included in the supplementary material in Section S7 and repeated here for convenience:
> >The proposed method will have positive societal impacts by enabling transparency while obtaining similar performance to the black-box counterparts. Nonetheless, it is still susceptible to adversarial attacks [1] similar to other existing deep learning models. A thorough inspection of the class-prototypes is thus required before it can be leveraged in safety critical scenarios to avoid spurious learning [2] or backdoor triggers [3].
>
> ---
>
> [1] Yansong et al. "Backdoor attacks and countermeasures on deep learning: A comprehensive review". ArXiv, abs/2007.10760, 2020.
>
> [2] Anders et al. "Analyzing imagenet with spectral relevance analysis: Towards imagenet un-hans’ed". ArXiv, abs/1912.11425, 2019.
>
> [3] Bryant et al. "Detecting backdoor attacks on deep neural networks by
> activation clustering". In SafeAI@AAAI, 2019.

---

> ### Author Response · Authors · 2022-08-09
> **Response to Reviewer iezD Part 2**
>
> We would like to further comment on the applicability of our method to real-world complex tasks.
>
> We do understand the concern that the reviewers raise with regards to the real-world applicability of our method and have therefore included additional results on the CelebA dataset in the paper. CelebA is another common real-world image generation dataset consisting of celebrity images, which has a considerably higher resolution (178×218) than any of the previously included datasets. The results and a discussion are included in Section S6.10 in the supplementary material, highlighting the quality of the prototypes that our model learns and showcasing its applicability to higher-resolution real-world datasets.
>
> We have in the past week further tried to get similar high-quality and crisp results for CIFAR-10, but unfortunately, CIFAR-10 can be considered quite a difficult dataset due to its high variation and low-resolution, which quickly leads to blurry image generation results. While recent generative approaches have shown promising performance on CIFAR-10, an extensive exploration of these backbones for our approach is infeasible given their multiple week long training runs (for Very Deep VAEs). We hope that the reviewer will consider the problem of getting crisp CIFAR-10 reconstructions out of the scope of this work, given the newly added results on another high-resolution real-world dataset.
>
> To further support our claim that the expressiveness of the backbone is crucial for reconstructing sharp prototypes, we have additionally added Section S6.11 in the supplementary material, which shows and discusses the test reconstructions for the MNIST, fMNIST, CIFAR-10, QuickDraw and SVHN datasets.
>
> We have updated the paper and supplementary material with the new results.

---

### Author Response · Authors · 2022-08-09
**General Response -- thanks to all reviewers for constructive and insightful feedback**

We would like to thank all the reviewers for appreciating our proposed methodology as well as the importance of the work, the predictive performance of our method, and our thorough evaluation.

Following the constructive suggestions and comments of the reviewers, we have revised our paper and included additional experimental results to highlight its applicability to complex real-world datasets. In particular, we have

* added a discussion on the scalability of ProtoVAE to more complex datasets;
* included qualitative results for the higher resolution real-world dataset CelebA;
* added additional baseline results on SVHN for more modern baselines;
* added an experiment that illustrates how increasing prototypes can reduce blurring effects;
* added missing error bars;
* and clarified among others the structure and reconstruction process of the prototypes as well as the LRP rules;

The main revisions in our paper and the supplemental material are marked in RED. We hope that our efforts address the concerns of all reviewers sufficiently.

---

### Meta-Review · Area_Chair_s3QW · 2022-08-26

**Recommendation:** Accept
**Confidence:** Certain

**Metareview:**

The paper is of good quality and presents an interesting approach to interpretability in a clear manner. Generally, the paper is very well written and due to extensive ablations and experiments offers many insights into the relevant aspects of the proposed approach. At the current stage the method might not be fully practical but might inspire further research down the line that overcomes said shortcomings. On the flip side, one drawback of the paper is relying on variational auto-encoder, which itself doesn't work nicely as a generative model for complex (natural) datasets. There are not many results on real-world complex tasks in the paper, and adding them would definitely add value. While there are some shortcomings, I believe it is still valuable for the research community. The authors have provided clarification to most major concerns in a convincing way. Overall, I suggest for the acceptance of the paper. Please improve the final version with the content from reviewer responses.

**Award:**

No

---

### Decision · Program_Chairs · 2022-09-14

Accept